# The Intrinsically Disordered Region in the Human STN1 OB-Fold Domain Is Important for Protecting Genome Stability

**DOI:** 10.3390/biology10100977

**Published:** 2021-09-28

**Authors:** Weihang Chai, Megan Chastain, Olga Shiva, Yuan Wang

**Affiliations:** 1Department of Cancer Biology, Cardinal Bernardin Cancer Center, Stritch School of Medicine, Loyola University of Chicago, Maywood, IL 60153, USA; 2Office of Research, Washington State University, Spokane, WA 99202, USA; megan.chastain@wsu.edu (M.C.); olga.shiva@wsu.edu (O.S.); yw682@cinj.rutgers.edu (Y.W.); 3Department of Radiation Oncology, Rutgers Cancer Institute of New Jersey, New Brunswick, NJ 08901, USA

**Keywords:** CST, replication stress, genome instability, STN1

## Abstract

**Simple Summary:**

The human CTC1–STN1–TEN1 (CST) complex is an ssDNA-binding protein complex that is thought to be related to the RPA70/RPA32/RPA14 complex. While recent studies have shown that CST plays key roles in multiple genome maintenance pathways, including protecting fork stability under perturbed replication, promoting efficient replication of difficult-to-replicate DNA, repairing DNA double-stranded breaks, and maintaining telomere integrity, it is poorly understood how CST function is regulated in genome maintenance. In this study, we identify an intrinsically disordered region (IDR) in the OB domain of STN1 and analyze the functions of cancer-associated IDR variants and a number of alanine substitutions of individual polar or hydrophilic residues in this IDR. We observe that these variants confer replication-associated genome instability, reduced cellular viability, and increased HU sensitivity. Analysis of protein–protein interactions using IDR variants and IDR deletion shows that the IDR is critical for STN1–POLα interaction, but not CST–RAD51 interaction or CST complex formation. Together, our results identify the IDR in STN1-OB as an important element modulating CST function in protecting genome stability under replication stress.

**Abstract:**

The mammalian CTC1–STN1–TEN1 (CST) complex is an ssDNA-binding protein complex that has emerged as an important player in protecting genome stability and preserving telomere integrity. Studies have shown that CST localizes at stalled replication forks and is critical for protecting the stability of nascent strand DNA. Recent cryo-EM analysis reveals that CST subunits possess multiple OB-fold domains that can form a decameric supercomplex. While considered to be RPA-like, CST acts distinctly from RPA to protect genome stability. Here, we report that while the OB domain of STN1 shares structural similarity with the OB domain of RPA32, the STN1-OB domain contains an intrinsically disordered region (IDR) that is important for maintaining genome stability under replication stress. Single mutations in multiple positions in this IDR, including cancer-associated mutations, cause genome instabilities that are elevated by replication stress and display reduced cellular viability and increased HU sensitivity. While IDR mutations do not impact CST complex formation or CST interaction with its binding partner RAD51, they diminish RAD51 foci formation when replication is perturbed. Interestingly, the IDR is critical for STN1–POLα interaction. Collectively, our results identify the STN1 IDR as an important element in regulating CST function in genome stability maintenance.

## 1. Introduction

Faithful duplication of genomic DNA during each cell cycle is crucial to ensuring that genetic material is accurately inherited by daughter cells. During DNA replication, replication forks often encounter a variety of barriers that stall replication fork progression. These barriers can arise from intrinsic factors such as DNA secondary structure formation, highly repetitive sequences, heterochromatic regions, and transcription-replication collision. In addition, various DNA damaging reagents such as ultraviolet radiation, DNA crosslinking agents, and DNA–protein crosslinking agents produce DNA lesions that block DNA polymerase progression [1,2]. Stalled forks need to be stabilized and protected from unscheduled nuclease attack and efficiently restarted to avoid genome instability.

The human CST complex, consisting of three subunits CTC1 (134 kD), STN1 (44 kD), TEN1 (14 kD), is an ssDNA-binding complex that resembles replication protein A (RPA) complex [3]. Mutations in CST genes give rise to two complex genetic diseases known as Coats plus and dyskeratosis congenita [4,5,6,7,8]. Originally identified as a binding partner of DNA polymerase α (POLα), CST assists POLα in priming and synthesizing DNA [9,10,11]. A number of studies have shown that CST is an important player in maintaining genome stability and telomere integrity [12]. Its telomeric functions include promoting efficient replication of telomeric DNA, preventing the accumulation of ssDNA in telomeric DNA, assisting POLα in synthesizing C-rich strands and thereby replenishing C-strand DNA, and restricting telomerase from excessively elongating telomeres [13,14,15,16]. In telomerase-negative cancer cells using the alternative telomere lengthening (ALT) pathway to maintain telomeres, CST localizes at APBs (ALT-associated PML bodies) and regulates C-circle production, although the mechanism underlying such regulation is unknown [17].

We have recently uncovered the important role of CST in maintaining the stability of nontelomeric sequences under replication stress. In response to hydroxyurea (HU)-induced replication stalling, CST forms microscopic foci that colocalize with RAD51 and is recruited to stalled replication forks [18]. CST facilitates RAD51 recruitment to forks stalled at GC-rich repetitive sequences and is important for the stability of these sequences [18]. In addition, CST binding to reversed forks can directly block MRE11 nuclease degradation of nascent strand DNA, thereby protecting fork stability [19].

The crystal structures of CST were first reported for STN1 and TEN1 [20], followed by the recent report on the cryo-EM structure of the decameric supercomplex formed by the whole CST complex in conjunction with the (TTAGGG)_3_ ssDNA [21]. CTC1, the largest subunit, contains seven oligonucleotide–oligosaccharide binding fold (OB-fold) domains (OB-A, B, C, D, E, F, G); STN1 contains one OB-fold domain at the N-terminus and two winged helix–turn–helix (wHTH) domains at the C-terminus; TEN1, the smallest subunit, forms one OB-fold domain [20,21]. These OB folds mediate CST binding to DNA and directly interact with CST-binding partners. CTC1 appears to be the major DNA-binding subunit, as residues in OB-F and OB-G domains are crucial for CST binding to telomeric DNA [21]. In addition, removing the N-terminal 700 amino acids of CTC1, which spans OB-A, OB-B, and OB-C domains, completely abolishes CST binding to DNA [3]. It has also been shown that the STN1 OB-fold domain is involved in DNA binding [22]. The CST complex formation involves the interaction between the OB-E of CTC1 and the wHTH1 domain of STN1 and the interaction between OB-G domain in CTC1, STN1-OB, and TEN1-OB [21]. The OB-B and OB-D domains of CTC1 and STN1-OB appear to participate in interacting with POLα [21,23].

While CST possesses multiple OB-fold domains like RPA and is considered RPA-like, its functions in genome protection pathways are distinct from RPA. Firstly, upon hydroxyurea (HU)-induced fork stalling, CST assists in RAD51 recruitment to stalled forks, whereas prior binding of RPA to ssDNA inhibits RAD51 filament formation [18,24,25]. Secondly, CST binding to DNA can directly block unscheduled MRE11 degradation of nascent strand DNA at reversed forks, while RPA binding to DNA has limited protection against MRE11 degradation [12]. At double-strand breaks (DSBs), CST interacts with the Shieldin complex and counters excessive end resection, thereby promoting canonical nonhomologous end-joining (c-NHEJ) [26,27]. In contrast, RPA binds to ssDNA at resected DSB ends to stimulate end resection by BLM–EXO1–DNA2, which is essential for promoting homologous recombination [28,29,30].

To better understand the molecular mechanism underlying how CST protects genome stability distinctly from RPA, we sought to analyze the structural elements of CST. We identify an intrinsically disordered region (IDR) in STN1-OB that is enriched in polar and charged residues. Unlike α-helices and β-sheets, IDRs fail to form stable structures, and they show high mobile flexibility. Despite being less known than organized structures such as α-helices and β-sheets, IDRs often mediate important protein–protein interactions and protein–DNA/RNA interactions and accommodate post-translational modifications [31,32,33]. To date, the Cancer Genomics database lists four cancer-associated STN1 alterations, namely missense mutations E95G, S96V, V97A, and S102T, residing within this region [34,35], indicating that this IDR may be important for STN1 function. In this report, we analyzed the functions of cancer-associated IDR variants E95G and S96V and a number of alanine substitutions of individual polar or hydrophilic residues in this IDR. We observe that these variants confer replication-associated genome instability, reduced cellular viability, and increased HU sensitivity. These variants do not affect the CST complex formation and have little impact on the CST–RAD51 interaction. Interestingly, they significantly impair HU-induced RAD51 foci formation. In addition, we also found that the IDR is critical for STN1–POLα interaction. Together, our results identify the IDR in STN1-OB as an important element modulating CST function in protecting genome stability under replication stress.

## 2. Materials and Methods

Cell lines and cell culture: HeLa, U2OS, and HEK293T cells were obtained from American Type Culture Collection. Cells were cultured in DMEM media supplemented with 10% cosmic calf serum (Hyclone, Logan, UT, USA, Thermo Fisher, Waltham, MA, USA) at 37 °C containing 5% CO_2_.

Plasmids: shRNA constructs pSIREN-shLUC (control shRNA targeting luciferase), pSIREN-shSTN1, RNAi-resistant pBabe-Flag-STN1, pCI-neo-Myc-STN1, pcDNA-HA-TEN1, and pcDNA-Flag-CTC1 were described previously [3,14,18,36]. All IDR mutations were generated by site-directed mutagenesis using the QuikChange II site-directed mutagenesis kit (Agilent, Santa Clara, CA, USA) and then sequenced to verify that no other mutations were introduced.

Antibodies: Primary antibodies: anti-STN1 (WB 1:1,000, Abcam, Cambridge, UK, ab89250); anti-Actin (WB 1:60,000, Sigma, St. Louis, MO, USA, A5441); anti-Flag (WB 1:2000, Sigma, St. Louis, MO, USA, F1804); anti-Myc (WB 1:500, Santa Cruz Biotechnology, Dallas, TX, USA, sc-40); anti-HA (WB 1:10,000, Bethyl, Montgomery, TX, USA, A190-108); anti-Flag (WB 1:4000, Millipore-Sigma, Burlington, MA, USA, F7245); anti-RAD51 (IF 1:10,000, Abcam, Cambridge, UK, ab63801); anti-POLA1 (WB, 1:2000, Bethyl, Montgomery, TX, USA, A302-851A).

Secondary antibodies: HRP goat anti-rabbit (WB 1:10,0000, Vector Laboratory, Burlingame, CA, USA, PI-1000); HRP goat anti-mouse (WB 1:5000, BD Pharmingen, San Diego, CA, USA, 554002); goat anti-mouse Alexa Fluor 488 (IF 1:1000, Thermo Fisher Scientific, Waltham, MA, USA, A11029); goat anti-rabbit Alexa Fluor 550 (IF 1:1000, Thermo Fisher Scientific, Waltham, MA, USA, 84541).

Co-immunoprecipitation (co-IP): HEK293T cells were co-transfected with Flag-CTC1, HA-TEN1, WT, or mutant Myc-STN1; treated with or without HU (2 mM) overnight; and collected. Cells were lysed in lysis buffer (0.1% NP-40, 50 mM Tris-HCl, pH 7.4, 50 mM NaCl, 2 mM DTT) supplemented with protease inhibitor cocktail, sonicated, and centrifuged at 13,000 rpm for 15 min at 4 °C. Supernatants were immunoprecipitated with anti-Myc antibody overnight at 4 °C with constant rotation. Beads were then washed three times with cold lysis buffer at 4 °C and then resuspended in lysis buffer with SDS sample loading buffer, boiled for 5 min, and subjected to Western blot analysis. Three independent co-IP experiments were performed for each mutant to ensure reproducibility.

Immunofluorescence staining (IF): IF was carried out as described previously [18], Briefly, cells were grown on coverslips or chamber slides and then fixed directly with 4% paraformaldehyde (PFA) in PBS for 15 min. Following fixation, cells were permeabilized with 0.15% Triton X-100 in PBS for 15 min, washed three times for 5 min with PBS, blocked with 10% BSA at 37 °C for 1 h in a humidified chamber, and then incubated with appropriate primary antibodies for overnight at 4 °C. Samples were then washed with PBS three times, incubated with secondary antibodies at room temperature for 1.5 h, and finally washed three times in PBS. Slides were then treated with cold ethanol series and dried in dark. Nuclei were visualized by counterstaining with DAPI mounting medium (Vector Laboratories, Burlingame, CA, USA). Z-stack images were obtained at a 0.3 µm thickness per slice under a Zeiss AxioImager M2 epifluorescence microscope (Carl Zeiss Microscopy, LLC, White Plains, NY, USA) with a 40× or 100× objective. Single Z-slice or max projection images were selected as representative images.

Chromosome breakage measurement: Chromosome fragmentation assay was performed as described [18]. To quantify chromosome breakages, metaphase images were obtained by Metasystem (MetaSystems Group, Inc., Medford, MA, USA) equipped with Zeiss AxioImager epifluorescence microscope with a 63× oil objective.

Colony formation assay: One hundred cells were seeded in 6-well plates in triplicate 2 days prior to treatment. Cells were then treated with 1 mM HU for 10 h, and HU was then removed and fresh medium was added to wells. After 10 days of incubation, the medium was removed, and cells were washed with PBS before being fixed and stained with the crystal violet solution (0.1% crystal violet, 1% methanol, and 1% formaldehyde). Cell viability was calculated by normalizing the colony numbers of the untreated control sample. Two independent biological replicates were performed.

## 3. Results

### 3.1. The aa 90–116 Region in STN1 Lacks a Defined Structure and Is Evolutionarily Conserved

The CST complex is considered RPA-like in that both complexes contain multiple OB-fold domains. RPA70, RPA32, and RPA14 contain four, one, and one OB domains, respectively, CTC1, STN1, and TEN1 contain seven, one, and one OB domains, respectively. Overlay of STN1-OB/TEN1 with RPA32-OB/RPA14 shows that these two structures can superimpose upon one another (Figure 1B). We notice that while the majority of the STN1 N-terminal residues participate in the OB-fold formation, a stretch of residues from 90 to 116 within this OB domain lacks a defined 3D structure (Figure 1A and the dashed line in Figure 1B) [20]. This region connects β-strands 3 and 4 and is rich in hydrophilic polar residues and charged residues, indicating that this region may be an IDR. Similarly, RPA32 also contains an IDR between β3 and β4 strands albeit with a shorter sequence [37] (Figure 1A). Since IDRs are often important regions regulating protein functions, we hypothesized that the IDR might be evolutionarily conserved. Analysis of STN1 sequences from various organisms reveals that this IDR sequence is highly conserved in higher eukaryotes (Figure 1C). The IDR sequence is enriched with polar and charged residues with a high isoelectric point (pI = 10) (Figure 1C), suggesting that it is hydrophilic and may be exposed to the surface.

### 3.2. Cancer-Associated Variants in the IDR Cause Chromosome Instabilities under Replication Stress

IDRs often play an important role in protein–protein interactions and protein–DNA/RNA interactions and also accommodate post-translational modifications [31,32]. Four cancer-associated variants reside in this IDR—E95G is found in breast carcinoma, S96V in cutaneous melanoma, V97A in cervical squamous cell carcinoma, and S102 in uterine endometrioid carcinoma [34,35]. Given that CST plays an important role in maintaining genome stability under replication stress, we tested the effects of two variants E95G and S96V on chromosome stability. We stably expressed the RNAi-resistant Flag-tagged E95G and S96V variants in HeLa cells by retroviral transduction and concurrently depleted endogenous STN1 using shRNA (Figure 2A). Cells were then treated with or without hydroxyurea (HU), and chromosome instabilities were measured. Consistent with our previous report [18], STN1 depletion induces spontaneous chromosome abnormalities including chromosome breaks and radial chromosomes, which are elevated by HU treatment (Figure 2B,C). While chromosome abnormalities induced by STN1 depletion were fully rescued by expression of the RNAi-resistant WT-STN1, E95G or S96V failed to rescue (Figure 2B,C), indicating that cancer-associated variants are deficient in maintaining genome stability. 

### 3.3. Polar Residues in the IDR Are Important for Maintaining Chromosome Stability

Since the STN1 IDR is rich in polar residues such as serines/threonines that are largely conserved among different species (Figure 1C), we hypothesized that at least some serines or threonines might be important for STN1 molecular function. Among all serine and threonine residues in this IDR, T94 and S98 are in close proximity to the two cancer-associated mutations E95G and S96V, while S111A is a putative ATM/ATR phosphorylation site (SQ). We then changed them individually to the nonpolar residue alanine, expressed the RNAi-resistant nonpolar variants in HeLa cells by retroviral transduction, and concurrently depleted endogenous STN1 using shRNA (Figure 2D). As shown in Figure 2E, all three mutations failed to rescue chromosome instabilities caused by STN1 knockdown. Thus, at least five residues in the IDR (T94, E95, S96, S98, S111) are critical for genome maintenance.

### 3.4. IDR Variants Impair Cell Viability and Have Increased Sensitivity to HU Treatments

We then analyzed the effect of IDR variants on cell viability under both unperturbed and perturbed replication conditions using colony formation assays. As shown in Figure 3, STN1 suppression markedly reduced cell proliferation under both unperturbed and HU-treated conditions, consistent with STN1’s role in protecting fork stability and promoting replication under replication stress. While WT-STN1 fully rescued such proliferation defects, all of the IDR variants were unable to rescue proliferation defects caused by STN1 suppression.

### 3.5. Effects of IDR Mutations on RAD51 Foci Formation under Replication Stress

RAD51 is the major player in stabilizing stalled forks and restarting stalled replication. In response to replication stress, RAD51 is recruited to stalled forks and mediates fork reversal to protect fork stability [38]. In addition, RAD51 antagonizes unscheduled nascent strand degradation caused by nucleases [39,40,41,42,43,44,45]. Previously we have observed that downregulation of CST diminishes RAD51 foci formation after HU treatment and attenuates the recruitment of RAD51 to stalled sites after HU treatment [18], suggesting that CST facilitates RAD51 recruitment to stalled forks. We therefore tested the effects of IDR mutations on HU-induced RAD51 foci formation. We found that none of the IDR mutations rescued RAD51 foci formation caused by STN1 depletion (Figure 4), indicating that the IDR is important for modulating RAD51 recruitment to stalled sites.

### 3.6. IDR Mutations Retain the CST Complex Formation and RAD51–CST Interaction

Next, to understand the molecular functions of IDR in modulating STN1 function, we attempted to determine the mechanism underlying the attenuated RAD51 foci formation caused by IDR mutations. Since the DNA binding ability of CST is primarily mediated by OB domains in CTC1 [21], we focused on protein–protein interactions. We have previously found that CST physically interacts with RAD51 in response to HU treatment, and we proposed that the CST complex helps in recruiting RAD51 to stalled forks via interacting with RAD51 [18]. We then analyzed whether IDR mutations were defective in CST complex formation and RAD51 interaction. Cells were co-transfected with Flag-CTC1, Myc-STN1 (WT and various IDR mutants), and HA-TEN1 and treated with HU. Co-IP was then performed with the anti-Myc antibody, and the precipitates were analyzed for the presence of Flag-CTC1, HA-TEN1, and RAD51. We found that all IDR STN1 mutants were able to form a complex with CTC1 and TEN1 just like WT-STN1 (Figure 5A), suggesting that the IDR has an insignificant role in CST subunit interactions. It has been reported that changing STN1 residues 110–121 (TSQLKKLQETIE) to NAAIRSNAAIRS does not affect CST complex formation [21]. Thus, it is unlikely that the IDR and the surrounding residues participate in CST complex formation. In addition, co-IP also showed that the IDR mutations had little impact on RAD51 interaction (Figure 5A). Collectively, our results indicate that the IDR is likely dispensable for interacting with CTC1–TEN1 or RAD51, and the IDR may modulate RAD51 recruitment to forks in a manner independent of CST–RAD51 interaction.

### 3.7. The IDR Modulates CST–POLα Interaction

Since CST is an accessory factor of POLα and stimulates POLα activity by promoting RNA priming and primase-to-polymerase switch [9,10], we next examined whether the IDR mutations affect CST interacting with POLα. We found that while WT-STN1 was able to pull down POLα in both untreated and HU-treated cells, T94A, S111A, and E95G impaired POLα interaction (Figure 5B), suggesting that the IDR may participate in interacting with POLα. Since the impact of individual mutations on STN1–POLα interaction was mild, we then deleted the entire 26 residues of the IDR (ΔIDR) and performed co-IP. Like individual IDR single mutations, the CST complex formation was largely unaffected by ΔIDR (Figure 5B). Since STN1-OB is essential for interacting with CTC1 and forming the CST complex, this observation suggests that removing the IDR has little impact on the proper folding of the overall OB-fold structure. Interestingly, ΔIDR markedly reduced STN1–POLα interaction, suggesting that IDR may contain an interface critical for CST–POLα interaction (Figure 5B). Notably, ΔIDR did not completely abolish the interaction, likely due to the participation of other POLα-interacting regions in the CTC1 OB-B and OB-D domains [21]. It remains to be determined in which way this IDR participates in POLα interaction and how disrupting such interaction impairs genome maintenance.

## 4. Discussion

The human CST trimeric complex has emerged as an important constituent that ensures faithful replication of genomic DNA, especially when replication is perturbed. It localizes at stalled replication forks and protects nascent strand DNA from being degraded by unscheduled nuclease degradation [12]. In this study, we report that the OB-fold structure of STN1–TEN1 is strikingly similar to that of RPA32–RPA14 (Figure 1B). We identify an IDR connecting the β3 and β4 strands of STN1-OB that is characterized by a high pI value and enrichment in polar and charged residues and participates in STN1–POLα interaction. We show that substituting selected polar and charged residues to nonpolar residues (alanine, glycine, valine) in the IDR induces replication-associated genome instabilities and impairs cell survival. Thus, this IDR is a critical structural element in regulating CST function in protecting genome stability during replication. Moreover, two cancer-associated alterations residing in the IDR display increased genome instabilities, suggesting that genome instabilities caused by the IDR alterations may play a role in tumorigenesis.

Our results show that amino acid alterations in the IDR attenuate HU-induced RAD51 foci formation (Figure 4). Previously we have found that CST physically interacts with RAD51, and CST deficiency impairs RAD51 foci formation, leading us to propose that CST binds to ssDNA formed at stalled forks and facilitates RAD51 recruitment to stalled forks via interacting with RAD51 [18]. Surprisingly, IDR variants have no obvious impact on CST–RAD51 interaction, suggesting that this IDR may regulate RAD51 filament formation via a mechanism independent of CST–RAD51 interaction. Using the deletion mutant with the entire IDR removed, our co-IP results show that the IDR region is crucial for CST interaction with POLα, reinforcing the importance of CST–POLα interaction in protecting genome stability. A recent report shows that CST nuclear localization is dependent on POLα interaction [46]. However, the IDR variants display normal nuclear localization (data not shown). Thus, it is possible that the CST–POLα interaction likely regulates CST and POLα function in genome replication via other unidentified mechanisms. Given that the IDR is rich in serine/threonine residues, we speculate that residues in this IDR may be targeted by post-translational modifications (PTMs). Although this work does not pinpoint the PTM site(s) or pathways, it is noticed that S111 is a putative ATR phosphorylation site. Since ATR plays a key role in responding to DNA replication stress, it will be interesting to investigate the potential role of ATR-mediated phosphorylation in modulating CST function at stalled forks. Additionally, the multiple serine/threonine residues in the IDR may be modified by other kinases in response to fork perturbation. Further investigation is needed to determine the PTM mechanisms.

While it is considered RPA-like, CST possesses functions distinct from RPA in fork protection and DSB repair. At stalled forks, CST binding to DNA prevents unscheduled MRE11 degradation of nascent strand DNA, while RPA has limited protection against MRE11 [12]. CST facilitates RAD51 recruitment to forks, whereas RPA-coated ssDNA inhibits RAD51 nuclear filament formation [18]. During DSB repair, CST appears to promote c-NHEJ via inhibiting excessive end resection through a POLα-dependent fill-in mechanism [26,27,47], whereas RPA stimulates end resection and promotes homologous recombination [28,29,30]. The structural difference between CST and RPA may help in explaining distinct functionalities of these two ssDNA-binding complexes in various genome maintenance pathways. While RPA32 also contains an IDR (Figure 1A), its IDR sequence shares no obvious homology with the STN1 IDR. We propose that the STN1 IDR plays an important role in regulating the unique function of CST during DNA replication, likely through interacting with POLα or other CST binding partners.

## 5. Conclusions

In this study, we find that substituting polar and charged residues to nonpolar residues in the IDR induces replication-associated genome instabilities and impairs cell survival. Such amino acid substitutions attenuate replication stress-induced RAD51 foci formation. Interestingly, the IDR mutants have little impact on CST complex formation and RAD51 interaction. We also find that the IDR is critical for STN1 interacting with POLα. Taken together, our findings uncover that the IDR in STN1-OB plays an important role in regulating CST function in protecting genome stability. Further investigation is needed to understand the molecular mechanisms regulating CST functions in protecting genome stability under replication stress via this IDR. 

## Figures and Tables

**Figure 1 biology-10-00977-f001:**
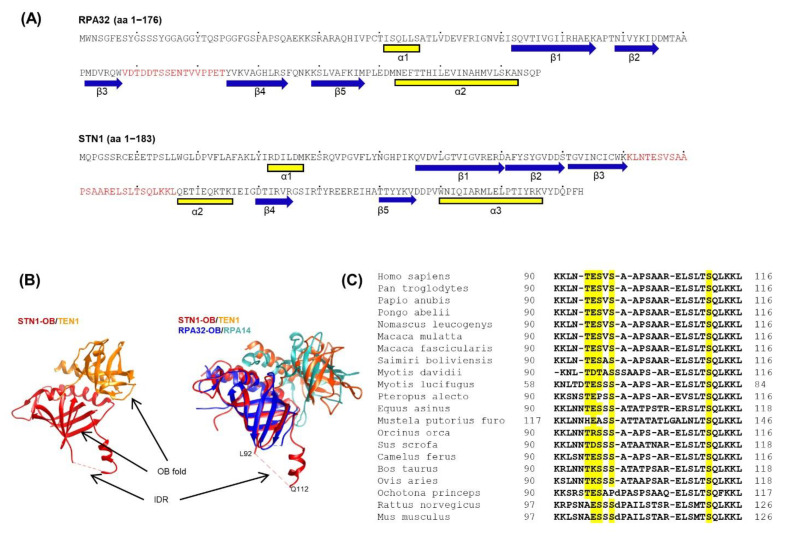
STN1 IDR position and conservation among different species. (**A**) Annotated protein sequences and secondary structures of the N-termini of RPA32 and STN1 containing the OB domains. α-helices are yellow. β-strands are blue. Residues in IDRs between β3/β4 of RPA32-OB and STN1-OB are red. Dots above the sequence indicate residue positions at an interval of 10. (**B**) Crystal structure of STN1-OB domain and TEN1 and overlay of STN1-OB/TEN1 with RPA32-OB/RPA14. Structures are derived from Protein Data Bank (PDB) with structure code 4JOI. RPA structures are derived from structure code 2PI2 in PDB. (**C**) Sequence of STN1 IDR in different species. Residues analyzed in this study (T94, E95, S96, S98, S111) are highlighted.

**Figure 2 biology-10-00977-f002:**
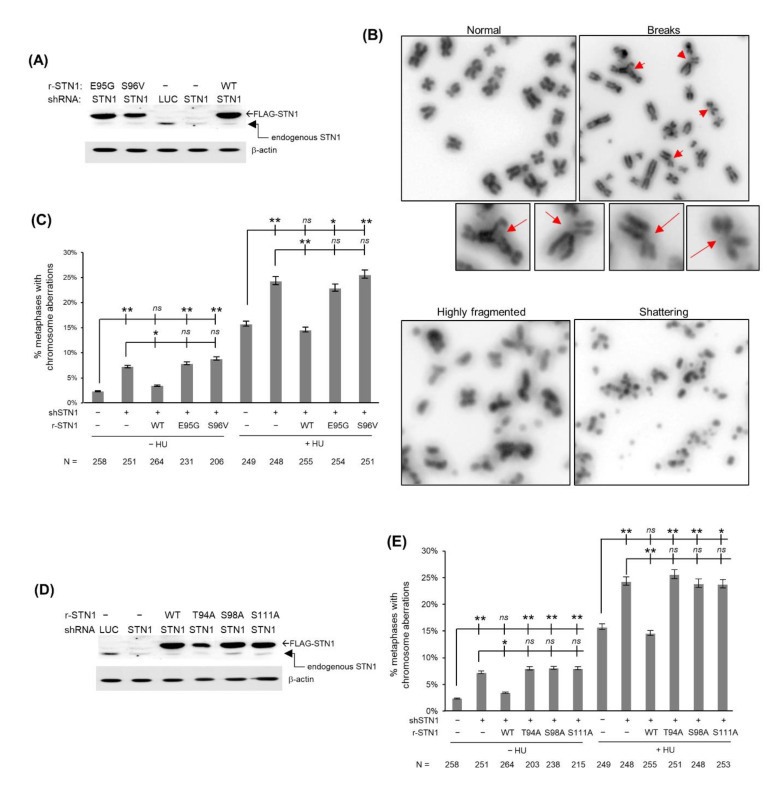
The IDR is important for maintaining chromosome stability. (**A**) Stable expression of RNAi-resistant (r-STN1) cancer-associated variants Flag-E95G and Flag-S96V. Endogenous STN1 was depleted with shRNA. Full Western blot images are provided in Appendix A. (**B**) Representative images of chromosome aberrations in STN1 knockdown cells concurrently expressing WT-STN1 or IDR variants. (**C**,**E**) Percent of metaphase spreads containing chromosome aberrations. Statistical analysis was performed with one-way ANOVA. N: the number of metaphase spreads analyzed in each sample. Error bars: SEM. ** *p* < 0.01, * *p* < 0.05. (**D**) Stable expression of RNAi-resistant Flag-T94A, Flag-S98A, and Flag-S111A with concurrent depletion of endogenous STN1.

**Figure 3 biology-10-00977-f003:**
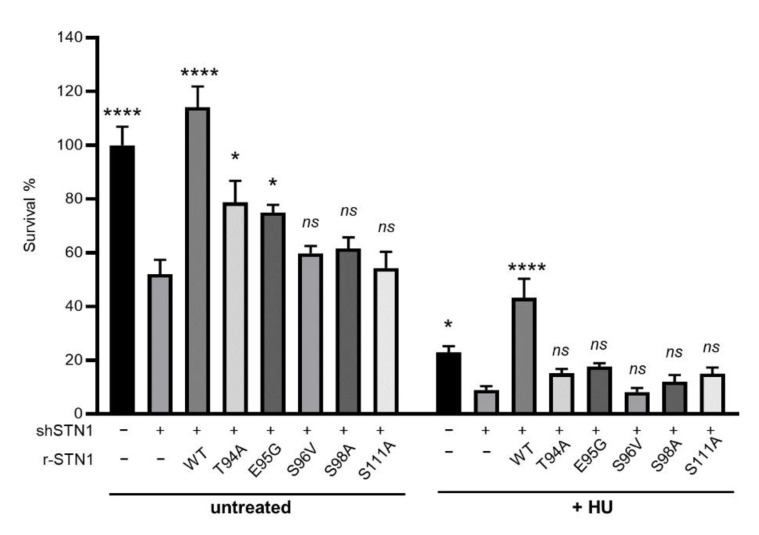
STN1 IDR variants impair cell proliferation. Colony formation results were from 2 independent experiments with triplicate in each experiment. Each value is normalized to the mean of untreated control knockdown. Error bars: SEM. One-way ANOVA was performed by comparing to shSTN1 to calculate statistical significance. **** *p* < 0.0001, * *p* < 0.05.

**Figure 4 biology-10-00977-f004:**
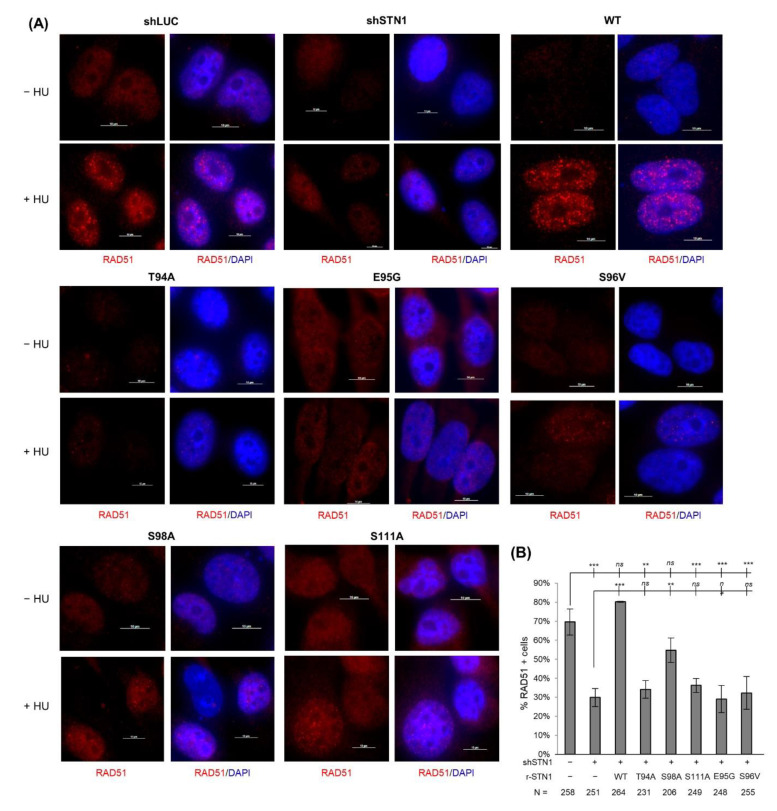
STN1 IDR mutations impair HU-induced RAD51 foci formation. (**A**) Representative images of RAD51 foci formation from HeLa cells stably expressing RNAi-resistant WT and IDR variants. Endogenous STN1 was concurrently depleted with shRNA. (**B**) Percent of RAD51 foci positive cells. Results were means from four independent experiments. At least 150 cells were analyzed for each sample. Error bars: SEM. One-way ANOVA with post hoc Tukey was performed by comparing to shSTN1 to calculate statistical significance. *** *p* < 0.001, ** *p* < 0.01.

**Figure 5 biology-10-00977-f005:**
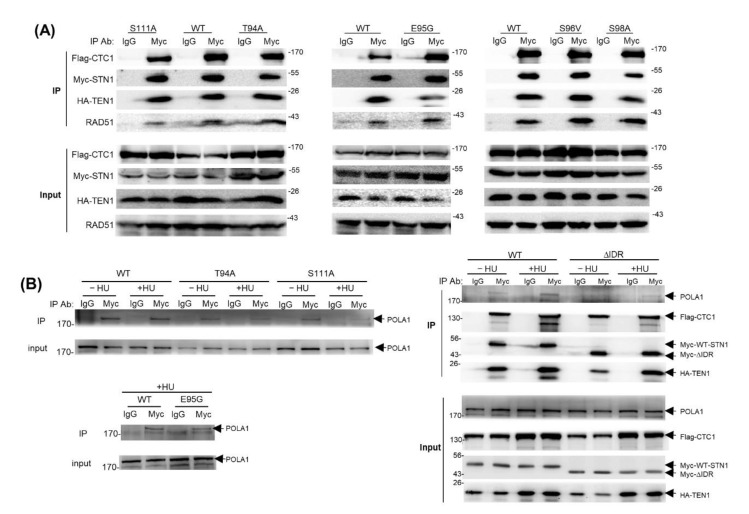
Impact of STN1 IDR mutations on CST complex formation, CST interaction with RAD51 and POLα. (**A**) Co-IP of STN1 IDR variants with CTC1, TEN1, and RAD51. HEK293T cells were co-transfected with Flag-CTC1, Myc-STN1, and HA-TEN1 and treated with HU (2 mM, 16 h), and co-IP was performed with anti-Myc antibody. Three independent co-IP experiments were performed for each mutant to ensure reproducibility. Representative results are shown. (**B**) Co-IP of STN1 IDR variants with POLA1. HEK293T cells were co-transfected with Flag-CTC1, Myc-STN1, and HA-TEN1 and treated with or without HU (2 mM, 24 h), and co-IP was performed with anti-Myc antibody. Anti-POLA1 was used to detect POLA1 in precipitates. Two independent co-IP experiments were performed for each mutant to ensure reproducibility.

## Data Availability

Not applicable.

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
