# Peer review of "The Intrinsically Disordered Region in the Human STN1 OB-Fold Domain Is Important for Protecting Genome Stability"

_biology, 2021, doi:10.3390/biology10100977_

Round 1

Reviewer 1 Report

This manuscript Biology-1373580 reported the discovery of the IDR region of STN1 from the human CST complex. The topic is a hot area in the general fields of DNA replication/ repair and telomere maintenance. The biology phenotype is solid and convincing, and the biochemistry/cell biology part needs small improvement before final acceptance.

  1. Please revise the pictures provided in Figure 2B into a zoomed view. It is one of the most solid phenotypic data in this manuscript and the authors seemed not realize so and provided in very small size. I suggest enlarging the pictures by 2-3 folds for both height and width.

  1. In Figure 5B, the POLA1 input for T94A mutant under both conditions, as well as the +HU group from S111A mutant, seemed to be very concerning to my eyes. They looked to me intentionally tuned down to obtain the conclusion that the interactions were attenuated through T94 and S111 sites. Please re-do this Figure 5B to make sure all loading controls for POLA1 are normalized equally and along with the revised figure, please quantify the western blog gel bands, using input to normalize it.

I also have an alternative suggestion for this part: remove the whole IDR region from the CST complex and evaluate the situation of the interaction between CST and POLA. This will provide a much cleaner view, in comparison to individual single mutant.

  1. Line 280, change “RNA-like” to “RPA-like”.

  1. By comparing STN1 and RPA32 structurally, in STN1 there is the region from A102 to T124 that can resemble the alpha-helix from RPA32 for trimerization between RPA hetero-trimer. Is it possible that among the mutants the authors tested, there are two types: a. POLA1-interaction deficient mutants; b. CST complex formation mutants? I believe the authors might want to discuss this aspect more carefully in the discussion because this field is highly competitive and sensitive.

Reviewer 2 Report

Weihang Chai et al., identified STN1 IDR (intrinsic disordered region) which plays critical role in mediating CTC1-STN1-TEN1 complex (CST) function in genome stability maintenance.

-The manuscript is well written.

-Mostly easy to follow and the figures are well put together.

-The quality of figures and table presented by authors are satisfactory.

- I find the manuscript satisfactory and recommend to accept it in its present form. 

Author Response

Thanks for the positive review.

Reviewer 3 Report

In this manuscript, the authors examine how mutations in an unstructured, intrinsically disordered region (IDR) of an OB-fold in STN1 impact genome integrity. The authors assay DNA breaks, cell viability, and Rad51 foci formation and find that multiple mutations (some of which are found in cancers) negatively impact the function of STN1. The authors' experiments are well-controlled, and their data are analyzed quantitatively. This work will be of interest to researchers studying CST function and thus I support the acceptance of this manuscript for publication.

My only minor comment is that the last two lanes (S98A and S11A) of the STN1 western blot in Figure 2D look like they may have been cropped from a different part of the membrane. If so, this should be stated in the figure legend.

Author Response

The reviewer is correct that Figure 2D is cropped from different parts of the same membrane in order to remove the samples in between. We have clarified this in the figure legend and also in the full western blot images provided in Supplemental information.